# Predictors of Health-Related Quality of Life in Korean Adults with Diabetes Mellitus

**DOI:** 10.3390/ijerph17239058

**Published:** 2020-12-04

**Authors:** Mihyun Jeong

**Affiliations:** Department of Nursing, Changshin University, Changwon 51352, Korea; mjeong@cs.ac.kr; Tel.: +82-55-250-3174

**Keywords:** health-related quality of life (HRQoL), diabetes mellitus (DM), Korean adults, Euro Quality of Life Five Dimension (EQ-5D)

## Abstract

Diabetes mellitus (DM) as a chronic disease is a major public health problem worldwide. It is important to improve the quality of life of people with DM, especially health-related aspects, which should be monitored and managed as part of diabetes management. Accordingly, this study investigated health-related quality of life (HRQoL) and identified predictors of HRQoL in Korean adults with DM using the Seventh Korea National Health and Nutrition Examination Survey (KNHANES VII) 2016–2018. This was a cross-sectional study with a stratified multistage probability sampling design that collected data from 1228 participants aged 30–80 years diagnosed with DM. HRQoL was measured using the Euro Quality of Life Five Dimension (EQ-5D) questionnaire. Analyses consisted of one-way analysis of variance, *t*-tests, chi-squared tests, and general linear regression analyses with complex sampling designs. Results demonstrated that low HRQoL was associated with older age (*β* = −0.002, 95% CI: −0.003 to −0.001), having elementary school education or less (*β* = −0.037, 95% CI: −0.061 to −0.014), being unmarried (*β* = −0.060, 95% CI: −0.078 to −0.040), poor subjective health status (*β* = −0.074, 95% CI: −0.094 to −0.055), perceived high stress (*β* = −0.047, 95% CI: −0.066 to −0.028), limited activity (*β* = −0.105, 95% CI: −0.131 to −0.079), being overweight (*β* = −0.021, 95% CI: −0.038 to −0.002), or obese (*β* = −0.016, 95% CI: −0.032 to −0.001), and three or more comorbidities (*β* = −0.044, 95% CI: −0.085 to −0.001). Comprehensive health care programs to manage these predictors should be provided to improve health-related quality of life of patients with DM.

## 1. Introduction

Diabetes mellitus (DM) as a chronic disease is a serious public health issue because the prevalence of diabetes has consistently increased worldwide. DM is a major cause of mortality and morbidity. Recently, the International Diabetes Federation (IDF) estimated that 629 million adults worldwide will have DM by 2045, which means the number of adults with diabetes will increase by 48% over the current number. In addition, approximately 50% of people who die from DM are less than 60 years old [1]. In Korea, the prevalence of diabetes in 2016 among adults aged 30 years or older was estimated to be 14.4%, and it was the sixth main cause of death [2].

Health-related quality of life (HRQoL) is a multidimensional concept that denotes the overall well-being and general health of individuals, which includes perceived physical and mental health, social and socioeconomic status, and community resources [3]. HRQoL is an important component of public health and is a strong predictor of mortality and morbidity [4,5]. In particular, as Korea rapidly progresses toward becoming an aging society, interest in, and demand for, health to promote improved quality of life is increasing.

It is important to assess the HRQoL of patients with diabetes that requires continuous medical care and self-management for glycemic control [6]. Diabetes management requires behavioral lifestyle modifications, such as regular self-monitoring of blood glucose, consistent physical activity, nutrition control, stress management, and adherence to diabetic medications because poor glycemic control causes microvascular complications such as retinopathy, neuropathy, and nephropathy, and increased risk of macrovascular complications, including stroke and heart diseases. These complications harm the quality of life of people with diabetes, especially in terms of health [7].

The HRQoL of people with diabetes is also affected by disease progression and concomitant diseases such as anxiety and depression, as well as physical and psychosocial health such as self-care and daily activities [8,9]. According to the guidelines of the American Diabetes Association (ADA), one of the goals of optimal diabetes management is to improve HRQoL [7]. This indicates the importance of assessing HRQoL as one outcome of diabetes.

Many previous studies of HRQoL in diabetes exist, and comprehensive reviews are available [9,10]. People with diabetes are more likely to experience low quality of life than the general population [6]. Poor HRQoL in people with diabetes is associated with adverse diabetes outcomes, including poor diabetes management [11]. Many studies have demonstrated that HRQoL is associated with age [12], sex [13], body mass index (BMI) [14], depression [15], physical activity, diabetes duration, comorbid conditions [16], and diabetes complications [17].

Therefore, this study examined HRQoL in Korean adults with diabetes and assessed differences in the subscales of the Euro Quality of Life Five Dimension (EQ-5D) in three groups defined by diabetes duration. We also identified predictors of HRQoL using a nationally representative sample of Korean persons with diabetes.

## 2. Materials and Methods

### 2.1. Data Source and Participants

This was a nationwide, cross-sectional study based on data from the Seventh Korea National Health and Nutrition Examination Survey (KNHANES VII) 2016–2018, which was performed by the Division of Chronic Disease Surveillance of the Korea Center for Disease Control and Prevention (KCDC) in the Ministry of Health and Welfare to investigate the general health and nutrition status of a representative sample of the Korean population. This sampling protocol employed a complex, stratified, multistage probability cluster sampling of the non-institutionalized civilian population of South Korea. Data were collected through a health examination, a health interview, and a nutrition survey by trained investigators via face-to-face interviews.

Of the 16,277 adults aged 19 years or older in the KNHANES VII (2016–2018), this study included participants who were aged 30–80 years, with self-reported DM, who completed the HRQoL questionnaire (*n* = 1438). The sample eligible for analysis consisted of 1228 participants, after further excluding participants with cancer (*n* = 59) and participants with missing data, including glycated hemoglobin (A1C) (*n* = 59), DM medication (*n* = 22), DM duration (*n* = 45), educational level (*n* = 5), perceived stress (*n* = 3), sleep duration (*n* = 10), household income (*n* = 4), physical activity (*n* = 1), smoking (*n* = 1), and subjective health status (*n* = 1). Figure 1 shows a flow chart diagram showing the selection of the study population.

Participants provided written informed consent before completing the KNHANES survey. The KNHANES is research that was directly conducted by the government for public welfare in accordance with the Korea Bioethics Act. The KNHANES VII (2016–2018) met the ethical standards of the National Research Committee, as well as the Helsinki Declaration-based ethical principles for all procedures involving human subjects. The data of the KNHANES VII provided are publicly accessible. The KNHANES VII was approved by the ethics review board of the KCDC (IRB No. 2018-01-03-P-A).

### 2.2. Instruments

#### 2.2.1. Sociodemographic Characteristics

Participants self-reported sociodemographic characteristics including age (years), sex (male or female), living area (urban or rural), educational level (elementary school or less, ≤6 years; middle school, ≤9 years; high school, ≤12 years; and college or more, >12 years), marital status (married or unmarried (including those who are single, divorced or widowed)), employment (yes or no), and household income (low, low-middle, middle-high, and high).

#### 2.2.2. Health-Related Characteristics

Current smoking, monthly drinking, and limited activity were treated as dichotomous variables (yes or no), as was perceived stress (low or high levels). In other words, current smoking was defined as smoking at least one cigarette per day in the past year and the presence of monthly drinking was defined as one or more drinks per month in the last year. Subjective health consisted of five conditions (very good, good, fair, poor, very poor) and was categorized as good, fair, or poor.

Physical activity was measured using the Global Physical Activity Questionnaire (GPAQ), which was developed by the World Health Organization to assess habitual physical activity during a typical week. The validity and reliability of the GPAQ has been established across Korea as well as in a variety of other countries [18]. The Korean version of the GPAQ was used for this study. The self-reported GPAQ consists of 16 items that assess three domains: work physical activity, transport physical activity, and recreational physical activity. Physical activity was classified as low or moderate-to-vigorous. Low physical activity was defined as 150 min or less of moderately intense exercise per week, or 75 min or less of highly intense exercise per week.

Self-reported sleep duration was defined as the average total nightly duration of sleep periods between sleep start and end and further categorized as short (5 h or less per night), normal (5 to 9 h per night), and long (9 h or more per night). These categories were selected based on the epidemiological literature regarding sleep duration among adults [19,20].

Body weight (kg) and height (cm) were measured by trained medical staff according to standardized procedures. BMI was calculated as body weight in kilograms divided by height in meters squared and categorized as non-obesity (<25 kg/m^2^) or obesity (≥25 kg/m^2^) using the criteria for the Asia-Pacifica region of the World Health Organization [21].

#### 2.2.3. Diabetes-Related Characteristics

Self-reported DM duration was categorized into 5 years or less, 6–10 years, or ≥11 years. DM treatment was divided into only oral hypoglycemic agents (OHA), only insulin, OHA and insulin, or no medication. Comorbidities related to DM included hypertension, dyslipidemia, stroke, myocardial infarction, and angina pectoris. Self-reported comorbid conditions were classified into four groups: none, 1, 2, or 3 or more. Glycated hemoglobin (A1C) was measured by Tosoh G8 high-performance liquid chromatography (Tosoh, Tokyo, Japan) through venous blood samples. A1C levels were categorized as good glycemic control (<7%) or poor glycemic control (≥7%) according to the American Diabetes Association (ADA) recommendations for glycemic target goals [7].

#### 2.2.4. Health-Related Quality of Life (HRQoL)

HRQoL was measured using the EQ-5D questionnaire, which is one of the most widely used measures of HRQoL. The instrument assesses five domains: mobility, self-care, usual activity, pain/discomfort, and anxiety/depression. Each domain of the EQ-5D has three response categories (1 = no problems, 2 = some problems, and 3 = severe problems). A unique health condition is determined by combining one level from each of the five domains of the EQ-5D, allowing for a total of 243 possible health conditions. These health conditions are converted into a single health index score that is calculated using the estimated weighted quality value for Koreans. The index scores range from −0.171 (severe problems were reported for all five EQ-5D health domains) to 1 (no problems were reported for all five EQ-5D domains), where negative scores are considered worse than death, 0 represents death, and 1 indicates perfect health. The validity and reliability of the Korean version of the EQ-5D has been established across diverse diseases [22,23,24].

### 2.3. Statistical Analysis

The data in this study were treated with consideration for stratification, sampling weights, and clustering according to the KNHANES data analysis guidelines. Complex sampling designs were used for all statistical analyses. Descriptive statistics were obtained to examine the background characteristics of participants and presented as frequencies and weighted percentages for categorical variables and means and standard errors for continuous variables. The Shapiro–Wilk test was applied to assess normality. One-way ANOVA of the complex sample’s general linear model was used to compare mean scores of HRQoL (EQ-5D) according to the background characteristics of the participants. Univariate linear regression analyses were performed to examine the associations between HRQoL (EQ-5D) and continuous variables including age and A1C. Chi-squared tests were used to compare each domain of the EQ-5D among three categories of diabetes duration (≤5 years, 6 to 10 years, and ≥11 years).

Complex-sample general linear regression analyses were performed to investigate factors that influence HRQoL (EQ-5D) in three models. Based on previous studies [14,25] and the outcomes of the Chi-squared tests and univariate linear regression analyses, Model 1 included sociodemographic factors of age, sex, educational level, marital status, employment, and household income. Health-related factors were added to Model 2, namely monthly drinking, subjective health status, perceived stress, limited activity, physical activity, sleep duration, and BMI. Finally, Model 3 included not only sociodemographic and health-related factors, but also diabetes-related factors, namely diabetes duration, diabetes medication, and comorbid chronic conditions. In multiple linear regression, multicollinearity between the independent variables was assessed by variance inflation factors (VIF). A VIF higher than 10 was considered as multicollinearity. A final multiple linear regression model was computed after checking that there was no multicollinearity. Statistical analysis was performed using SPSS version 22.0 (IBM Corp., Armonk, NY, USA). *p* values < 0.05 were considered statistically significant.

## 3. Results

### 3.1. Background Characteristics of Study Participants

Table 1 shows the background characteristics of participants with DM. The mean age of the 1228 participants was 63.33 ± 0.38 years, ranging from 30 to 80 years. There were more men (52.9%) than women. Slightly less than one-third had a high school education (30.3%) or elementary school or less education (31.7%). Approximately two-thirds (73.4%) of the participants were married and slightly more than half (51.6%) were working full or part time. More than a quarter of the participants (26.6%) reported a high household income level, while less than a quarter (23.7%) reported a low household income level.

In terms of health-related characteristics, only approximately one-fifth of participants (18.4%) were current smokers, whereas approximately half (46.0%) drank alcohol on a monthly basis. Slightly less than one half (48.1%) reported that their subjective health was fair, followed by poor (39.4%), and good (12.5%). Most participants (76.7%) perceived a low level of stress, and no activity limitations were reported by the majority (84.9%). Slightly less than two-thirds (63.7%) engaged in a low level of physical activity. The mean sleep duration was 7.17 ± 0.05 h per night and the majority of participants (79.7%) had a normal sleep duration of between 5 h and 9 h per night. Short sleep durations of 5 h or less were reported by only 9.2% (*n* = 114) of participants, and long sleep durations of 9 h or more were observed in 11.1% (*n* = 159) of the sample. The mean BMI was 25.22 ± 0.11 kg/m^2^. Participants with obesity (BMI ≥ 25 kg/m^2^) accounted for nearly half of the sample (49.3%), and approximately one-quarter (24.8%) were overweight (23 ≤ BMI < 25 kg/m^2^).

Regarding diabetes-related characteristics, participants had been diagnosed with diabetes by a physician for approximately 10 years on average, and similar proportions of participants had mean diabetes durations of five years or less (38.7%) and 11 years or more (38.2%). Most participants (92.1%) were prescribed oral medication for diabetes control, and only 6% (*n* = 79) used both medication and insulin. More than two-thirds had one or two comorbid conditions such as hypertension, dyslipidemia, stroke, myocardial infarction, and angina pectoris (41.3% and 32.8%, respectively). The mean glycosylated hemoglobin (A1C) level of participants was 7.20 ± 0.04%, and the mean HRQoL (EQ-5D) score was 0.91 ± 0.01. The EQ-5D scores, age, and A1C values were normally distributed.

### 3.2. Differences in HRQoL by Background Characteristics

Table 2 represents the differences in HRQoL according to the background characteristics of the participants. Self-reported HRQoL differed significantly according to age, sex, educational level, marital status, employment, household income, monthly drinking, subjective health status, perceived stress, limited activity, physical activity, sleep duration, DM duration, and comorbid conditions (*p* < 0.01). Differences in HRQoL were not significant by living area, current smoking status, BMI, DM medication, and A1C level (*p* > 0.05). The mean scores for HRQoL were significantly lower in persons with the following characteristics: an elementary school or lower level of education, being unmarried, low income, unemployed, poor subjective health status, high level of perceived stress, limited activity, low physical activity, and short sleep duration (*p* < 0.01). In addition, participants with longer diabetes duration or a greater number of comorbid health conditions had significantly lower mean HRQoL scores (*p* < 0.01).

### 3.3. Differences in EQ-5D Subscales by Diabetes Duration

Figure 2 shows the differences in the five domains of the EQ-5D according to three categories of diabetes duration. There were significant differences (*p* < 0.05) among the three different durations of diabetes in EQ-5D problems related to immobility, self-care, usual activities, and anxiety/depression. Participants with long diabetes durations of 11 years or more had higher percentages of problems with mobility (33.7%), self-care (11.4%), usual activities (21.8%), and anxiety/depression (14.8%) than those with a diabetes duration of five years or less, and between 6 and 10 years (*p* ≤ 0.05). Pain/discomfort was not different among the three categories (≤5 years, 6 to 10 years, and ≥11 years) of diabetes duration (*p* > 0.05).

### 3.4. Factors Influencing HRQoL

Table 3 shows the results of the complex-sample general linear regression analyses of factors associated with HRQoL. Significant negative predictors of HRQoL scores in Model 1, which included sociodemographic factors, were elementary education or less, being unmarried, unemployment, and middle-to-high and low-to-middle household income. These predictors accounted for 21% of the total variance. In Model 2, which added health-related factors, significant negative predictors of HRQoL scores were age, fair or poor subjective health status, high level of perceived stress, limited activity, and being overweight or obese. Elementary education or less and being unmarried remained significant predictors of HRQoL scores. These predictors accounted for 38% of the total variance. Finally, in Model 3, which added diabetes-related factors, significant negative predictors of HRQoL scores were no diabetes medication, and three or more comorbid conditions. Age, elementary education or less, being unmarried, fair or poor subjective health status, high level of perceived stress, limited activity, and being overweight or obese remained significant predictors of HRQoL scores. Model 3 accounted for 39% of the total variance.

## 4. Discussion

The present study identified factors that influenced HRQoL using data from KNHANES VII (2016–2018). This study used a nationally representative sample of South Korean adults with DM; therefore, the results are generalizable to all South Koreans with DM.

The findings showed that the mean HRQoL score of Korean adults aged 30–80 years with DM was 0.91, which is similar to the quality of life of people with chronic diseases, such as hypertension and chronic liver disease [26,27], as well as in general populations [28]. The KNHANES IV and V collected data from 2007 to 2012 and reported a mean HRQoL score of 0.90 in Koreans aged over 50 years [29]. In contrast, the mean HRQoL score in the current study was higher than that reported in previous studies of people with dementia [28] or cancer [30,31].

The findings of this study demonstrated that predictors of HRQoL in Korean adults aged 30–80 years with DM consisted of the sociodemographic factors of age, educational level, and marital status; the health-related factors of subjective health status, perceived stress, limited activity, and BMI; and the diabetes-related factors of diabetes medication and comorbid conditions. Sociodemographic and health-related factors explained most of the variance described by the model, whereas diabetes-related factors explained almost none of the variance in HRQoL. Lower HRQoL scores were reported by people with DM who were older, with elementary education or less, unmarried, with fair or poor subjective health status, a high level of perceived stress, limited activity, overweight or obese, no diabetes medication, and three or more comorbid conditions.

In particular, in this study, elementary school or lower education and being unmarried were strong predictors of worse HRQoL regardless of the addition of health-related factors and diabetes-related factors. Older age was associated with lower HRQoL due to factors related to health and diabetes. These findings are consistent with previous population-based studies in a variety of countries [28,32]. A public health study of the Swedish population reported an inverse relationship between age and EQ-5D index. The oldest Swedish subjects, aged 80 to 88 years, reported the lowest EQ-5D scores among all age groups [32]. A study of the relationship between demographic and socioeconomic factors and HRQoL in the U.S. general population reported that older age and lower levels of education were associated with lower EQ-5D scores [28]. A cross-sectional study of the Vietnamese elderly population aged 60 years or older with diabetes reported a negative relationship between old age and EQ-5D scores [33]. A community-based study that used the Short Form (SF)-36 health survey in Greece reported that decreased HRQoL in persons with type 2 diabetes was associated with unmarried status, older age, and lower education [34].

This study found that both poor psychological and physical conditions were associated with low HRQoL in patients with diabetes. These findings are partially consistent with those of previous studies. Most previous studies of HRQoL in people with diabetes reported that either psychological or physical factors affected HRQoL [6,14,17]. A cross-sectional study using the SF–36 in adults who were overweight or obese and had type 2 diabetes reported that low physical HRQoL was associated with high BMI and physical discomfort [17]. A study that used the EQ-5D to measure HRQoL in people with type 2 diabetes demonstrated that obesity was associated with lower HRQoL [14,25]. In contrast, a study using the KNHANES IV (2007–2009) data showed that psychological factors, such as depressive symptoms and subjective stress, influenced HRQoL [6]. A cross-sectional study of Jordanian patients with diabetic foot ulcers demonstrated that patients who reported a stressful life had significantly lower HRQoL according to the Diabetic Foot Scale Short Form and SF-8 summary scales [25]. Moeineslam et al. [35] reported that in women, diabetes had a negative effect on psychological quality of life. These health-related factors have an important effect on glycemic control in DM, which causes serious deterioration in not only HRQoL, but also general quality of life.

Of the diabetes-related factors, three or more comorbid conditions was significantly associated with lower HRQoL, but the effect size was small. Moreover, the longer the diabetes duration, the lower the HRQoL in the one-way ANOVA. However, there was no association between diabetes duration and HRQoL in the multivariate regression. A1C as an indicator of glycemic control did not appear to influence HRQoL. More interestingly, people with no diabetes medication reported lower HRQoL than those taking diabetes medications. A plausible reason is that people who were not taking diabetes medication might have already had a comorbid disease, such as stroke or hypertension [9,36]. These findings are partially consistent with previous cross-sectional studies regarding the relationship between diabetes-related factors and HRQoL [6,13,14,33,37]. In a Vietnamese study of people aged 60 years or older with type 2 diabetes, a significant decrease in HRQoL was associated with a longer duration of diabetes, use of insulin, and presence of comorbidity [33]. A Dutch study of people with type 2 diabetes showed that the most significant predictors of low HRQoL were insulin use and the presence of complications [14]. In a Greek study of people with 70 years as the mean age, diabetes duration and comorbidities such as hypertension and hyperlipidemia were associated with impaired HRQoL [37]. A study that used a self-administered quality of well-being index to assess HRQoL reported that major diabetes complications were related to worse HRQoL [13]. In a Korean study of people with diabetes, glycemic control was not correlated with low HRQoL, and persons with diabetes had a lower HRQoL than the general population without diabetes [6]. A Norwegian study of associations between diabetes complications and HRQoL reported that the strongest predictors of reduced HRQoL in people with diabetes were ischemic heart disease, stroke, and neuropathy [8]. A systemic review reported that comorbidities and diabetes complications influenced HRQoL in people with type 2 diabetes [9]. Therefore, research results are inconsistent regarding the relationship between HRQoL and diabetes duration, glycemic control, and diabetes medication.

In the subdomains of the EQ-5D used to assess HRQoL, this study reported that people with longer diabetes duration had more severe problems with mobility, self-care, and usual activities. As the duration of diabetes extends, diabetes complications, such as microvascular or macrovascular diseases, gradually develop, which can limit persons with DM in their usual activities and self-care in daily life. Eventually, overall quality of life may decline because of the presence of diabetes. This finding is consistent with previous studies that reported diabetes duration was associated with low HRQoL in terms of physical functioning [38,39].

The strength of the current study includes its large population, which was obtained via the recent KNHANES VII; the study therefore provides representative information on the Korean population. Nevertheless, there are several limitations that should be considered when interpreting this study. First, the cross-sectional design of this study restricts drawing causal relationships between HRQoL and potential predictors. Second, the presence of diabetes was self-reported; thus, the reliability of the diabetes diagnosis was uncertain. Third, this study focused on other comorbidities, including heart diseases (angina pectoris and myocardial infarction) and stroke, instead of assessment of diabetes complications because the KNHANES is not a survey only for diabetes, but rather a general survey of national health and nutrition conducted at the national level. Hence, further studies are needed to identify whether diabetes complications affect quality of life. Fourth, the KNHANES has annually used the EQ-5D, which assesses HRQoL for the general population [36]; thus, studies using diabetes-specific instruments are necessary to better assess the association between diabetes complications and HRQoL. Lastly, this study included data from patients with type 1 and type 2 diabetes. We could not identify the type of diabetes because diabetes was self-reported. Even though there is a negative impact on overall HRQoL in both types of DM, the influence on HRQoL may be different for type 1 and type 2 diabetes. Therefore, further studies are needed to investigate predictors of HRQoL according to the types of DM.

## 5. Conclusions

This study demonstrates that low HRQoL in diabetes is influenced by older age, elementary school education or less, unmarried status, poor subjective health status, highly perceived stress, limited activity, being overweight or obese, and three or more comorbidities. These predictors should be considered in the design and analysis of HRQoL estimations in diabetes. In particular, because subjective health status, perceived stress, BMI, and comorbid conditions are modifiable factors, comprehensive health care programs to manage these factors should be provided to promote better HRQoL in diabetes. In addition, future studies should examine the development and effectiveness of psychological intervention programs to improve HRQoL in patients with DM. The findings of this study suggest that combined approaches on psychological intervention and intensive diabetes management would be appropriate for HRQoL of patients with DM.

## Figures and Tables

**Figure 1 ijerph-17-09058-f001:**
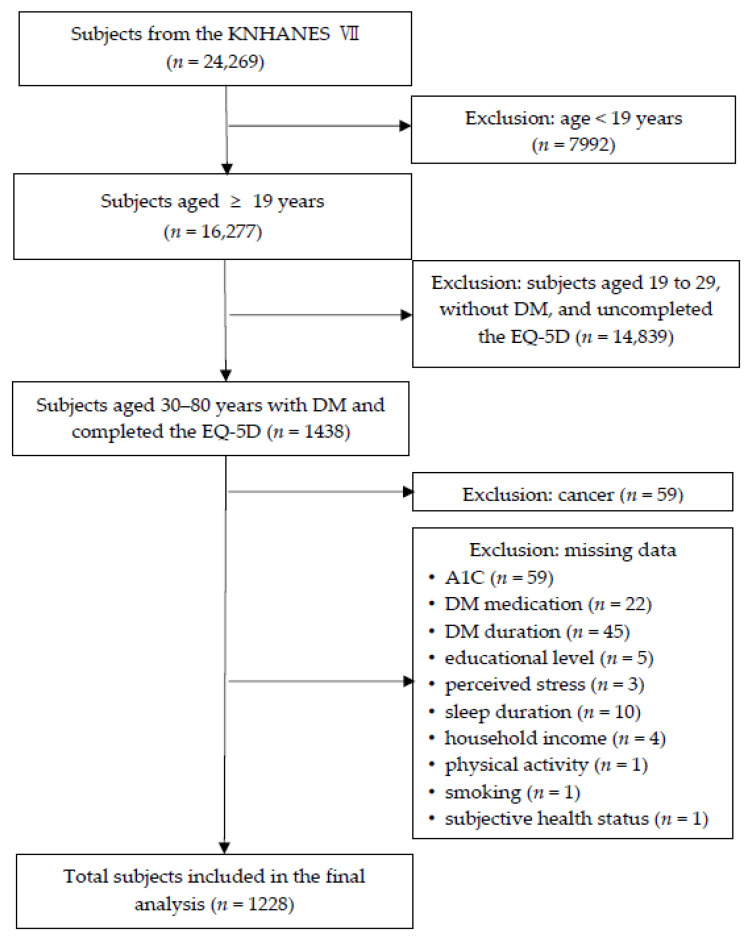
Flow chart diagram showing selection of the study population. KNHANES VII: Seventh Korea National Health and Nutrition Examination Survey; DM: diabetes mellitus; EQ-5D: Euro Quality of Life Five Dimension; A1C: glycated hemoglobin.

**Figure 2 ijerph-17-09058-f002:**
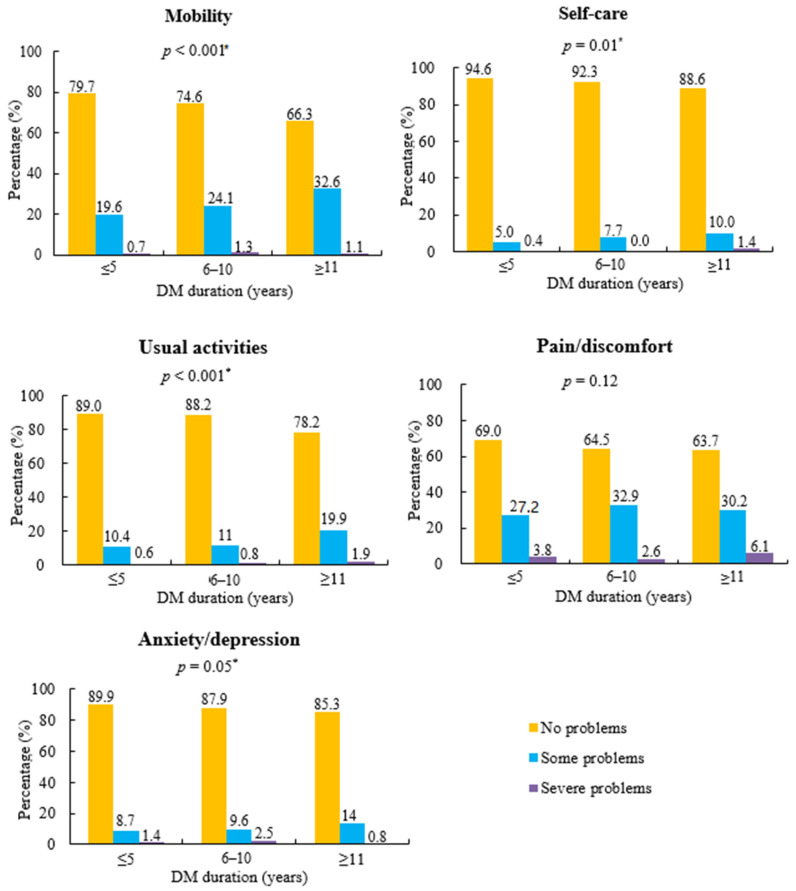
Distribution of perceived problem levels in each health domain of the EQ-5D by diabetes duration. * Significance level among three groups of diabetes mellitus (DM) duration by complex-sample chi-square test.

**Table 1 ijerph-17-09058-t001:** Background characteristics of study participants (*n* = 1228).

Characteristics	Distribution	*N* or M ± SE	Weighted %
**Sociodemographic Characteristics**			
Age, years		63.33 ± 0.38	
Sex	Male	613	52.9
	Female	615	47.1
Living area	Rural	259	17.3
	Urban	969	82.7
Educational level	College or more	192	18.6
	High school	336	30.3
	Middle school	247	19.4
	Elementary school or less	453	31.7
Marital status	Married	885	73.4
	Unmarried	343	26.6
Employment	Yes	570	51.6
	No	658	48.4
Household income	High	323	26.6
	Middle–high	311	24.5
	Low–middle	309	25.1
	Low	285	23.7
**Health-Related Characteristics**			
Currently smoking	Yes	203	18.4
	No	1025	81.6
Monthly drinking	Yes	514	46.0
	No	714	54.0
Subjective health status	Good	157	12.5
	Fair	580	48.1
	Poor	491	39.4
Perceived stress	Low	957	76.7
	High	271	23.3
Limited activity	Yes	198	15.1
	No	1030	84.9
Physical activity	Moderate-to-high	408	36.3
	Low	820	63.7
Sleep duration, h/night		7.17 ± 0.05	
	5 to 9 (normal)	955	79.7
	≤5 (short)	114	9.2
	≥9 (long)	159	11.1
BMI, kg/m^2^		25.22 ± 0.11	
	<18.5 (underweight)	13	1.0
	18.5 to 22.9 (normal)	311	24.9
	23.0 to 24.9 (overweight)	307	24.8
	≥25.0 (obese)	597	49.3
**Diabetes-Related Characteristics**			
DM duration, years		9.79 ± 0.27	
	≤5	435	38.7
	6 to 10	276	23.0
	≥11	517	38.2
DM medication	Only OHA	1127	92.1
	Only insulin	13	1.1
	OHA and insulin	79	6.0
	No medication	9	0.8
Comorbid conditions	0	206	18.5
	1	514	41.3
	2	413	32.8
	≥3	95	7.4
A1C, %		7.20 ± 0.04	
HRQoL (EQ-5D)		0.91 ± 0.01	

M: mean; SE: standard error; BMI: body mass index; DM: diabetes mellitus; OHA: oral hypoglycemic agent; A1C: glycated hemoglobin; HRQoL: health-related quality of life; EQ-5D = Euro Quality of Life Five Dimension.

**Table 2 ijerph-17-09058-t002:** Differences in HRQoL by background characteristics (*n* = 1228).

Variables	Mean ± SE	*p*-Value
**Sociodemographic Factors**		
Age, years	0.91 ± 0.00	<0.001
Sex		<0.001
Male	0.94 ± 0.01	
Female	0.87 ± 0.01	
Living area		0.566
Rural	0.90 ± 0.01	
Urban	0.91 ± 0.01	
Educational level		<0.001
College or more	0.96 ± 0.01	
High school	0.93 ± 0.01	
Middle school	0.92 ± 0.01	
Elementary school or less	0.84 ± 0.01	
Marital status		<0.001
Married	0.94 ± 0.00	
Unmarried	0.82 ± 0.01	
Employment		<0.001
Yes	0.94 ± 0.00	
No	0.87 ± 0.01	
Household income		0.001
High	0.93 ± 0.01	
Middle–high	0.92 ± 0.01	
Low–middle	0.90 ± 0.01	
Low	0.88 ± 0.01	
**Health-Related Factors**		
Currently smoking		0.419
Yes	0.91 ± 0.01	
No	0.90 ± 0.01	
Monthly drinking		<0.001
Yes	0.93 ± 0.01	
No	0.89 ± 0.01	
Subjective health status		<0.001
Good	0.97 ± 0.01	
Fair	0.94 ± 0.00	
Poor	0.84 ± 0.01	
Perceived stress		<0.001
Low	0.92 ± 0.00	
High	0.85 ± 0.01	
Limited activity		<0.001
Yes	0.77 ± 0.02	
No	0.93 ± 0.00	
Physical activity		<0.001
Moderate-to-high	0.93 ± 0.01	
Low	0.89 ± 0.01	
Sleep duration, h/night		0.007
>5 and <9 (normal)	0.91 ± 0.01	
≤5 (short)	0.87 ± 0.02	
≥9 (long)	0.88 ± 0.01	
BMI, kg/m^2^		0.077
<18.5 (underweight)	0.89 ± 0.04	
18.5–22.9 (normal)	0.92 ± 0.01	
23.0–24.9 (overweight)	0.91 ± 0.01	
≥25.0 (obese)	0.90 ± 0.01	
**Diabetes-Related Factors**		
DM duration, years		<0.001
≤5	0.92 ± 0.01	
6–10	0.91 ± 0.01	
>11	0.89 ± 0.01	
DM medication		0.086
Only OHA	0.91 ± 0.00	
Only insulin	0.93 ± 0.04	
OHA and insulin	0.85 ± 0.03	
No medication	0.87 ± 0.04	
Comorbid conditions		<0.001
0	0.93 ± 0.01	
1	0.92 ± 0.01	
2	0.89 ± 0.01	
≥3	0.82 ± 0.02	
A1C, %	0.91 ± 0.00	0.881

**Table 3 ijerph-17-09058-t003:** Complex-sample general linear regression analyses predicting HRQoL in people with DM (*n* = 1228).

Variables	Model 1		Model 2		Model 3	
	*β*	S.E.	*β*	S.E.	*β*	S.E.
**Sociodemographic Factors**						
Age, years	−0.001	0.001	−0.002 ***	0.001	−0.002 ***	0.001
Sex						
Male	1		1		1	
Female	0.011	0.008	0.006	0.008	0.009	0.008
Educational level						
College or more	1		1		1	
High school	−0.013	0.008	−0.013	0.007	−0.014	0.008
Middle school	−0.014	0.010	0.001	0.009	0.000	0.010
Elementary school or less	−0.061 ***	0.014	−0.037 **	0.012	−0.037 **	0.012
Marital status						
Married	1		1		1	
Unmarried	−0.072 ***	0.012	−0.059 ***	0.010	−0.060 ***	0.010
Employment						
Yes	1		1		1	
No	−0.032 ***	0.010	−0.013	0.008	−0.009	0.008
Household income						
High	1		1		1	
Middle–high	−0.012 **	0.010	0.003	0.009	0.002	0.009
Low–middle	−0.026 *	0.011	−0.008	0.010	−0.007	0.010
Low	−0.029	0.012	0.001	0.011	0.001	0.011
**Health-Related Factors**						
Monthly drinking						
Yes			1		1	
No			0.006	0.008	0.007	0.008
Subjective health status						
Good			1		1	
Fair			−0.023 ***	0.007	−0.022 **	0.007
Poor			−0.080 ***	0.010	−0.074 ***	0.010
Perceived stress						
Low			1		1	
High			−0.047 ***	0.010	−0.047 ***	0.010
Limited activity						
Yes			−0.107 ***	0.014	−0.105 ***	0.013
No			1		1	
Physical activity						
Moderate-to-high			1		1	
Low			−0.002	0.007	−0.002	0.007
Sleep duration, h/night						
>5 and <9 (normal)			1		1	
≤5 (short)			−0.018	0.015	−0.021	0.015
≥9 (long)			0.008	0.011	0.007	0.011
BMI, kg/m^2^						
<18.5 (underweight)			−0.001	0.029	−0.006	0.029
18.5–22.9 (normal)			1		1	
23.0–24.9 (overweight)			−0.021 *	0.009	−0.021 *	0.009
≥25.0 (obese)			−0.018 *	0.008	−0.016 *	0.008
**Diabetes-Related Factors**						
DM duration, years						
≤5					1	
6–10					−0.004	0.009
>11					−0.002	0.009
DM medication						
Only OHA					1	
Only insulin					0.023	0.024
OHA and insulin					−0.024	0.019
No medication					−0.070 *	0.032
Comorbid chronic condition						
0					1	
1					0.007	0.009
2					−0.004	0.010
≥3					−0.044 *	0.021
*R^2^*	0.21		0.38		0.39	

Note: * *p* < 0.05, ** *p* < 0.01, *** *p* < 0.001.

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
