# Peer review of "Predictors of Health-Related Quality of Life in Korean Adults with Diabetes Mellitus"

_ijerph, 2020, doi:10.3390/ijerph17239058_

Round 1
Reviewer 1 Report
Review of the manuscript entitled “Predictors of Health-related Quality of Life in Korean Adults with Diabetes Mellitus”
Journal: International Journal of Environmental Research and Public Health.
Dear Editor,
Thank you for the possibility of peer-reviewing the article entitled “Predictors of Health-related Quality of Life in Korean Adults with Diabetes Mellitus”. The article is a cross-sectional study investigating the health-related quality of life in Korean adults with Diabetes Mellitus as well as the differences in the five domains of the Euro Quality of Life-5 Dimension (EQ-5D) in three groups specified by the duration of diabetes. The study presents some interesting insights, such as the focus on the predictors of health-related quality of life in a large sample of people with diabetes; however, to be considered for publication, in my opinion, the manuscript needs several major and minor revisions.
Abstract
The abstract is well-written, and it is very clear to the reader. However, I suggest the Authors add the range age of the sample to allow a clearer understanding of the sample.
Data source and Participants
Also, here, I suggest the Authors add the range age of the sample. Indeed, in the Discussion section, the Authors use the word “elderly” or “older people”, and this creates a little bit of confusion since the sample comprises adults. Please better clarify it. In this section, there is a typo, quoting the Authors “pregnant women (n = 0)”. In other words: 1438 have completed the questionnaire, 1228 are the actual participants, but to reach this number 210 subjects have been excluded, here 4 subjects are missing. Please, correct this aspect.
Statistical Analysis
Authors should add whether the sample is distributed normally or not, using the Shapiro-Wilk test, thereby add an explanation of the normal distribution at the beginning of the Results section. Moreover, the Authors should be more concise in their explanation of statistical analyses since they repeat several times the same concept. I do not understand the reason why the Authors said that they carried out the one-way ANOVA and then the t-test: they are the same thing. Table 2 shows the one-way ANOVA, and I did not find the t-test analyses. Please, be more rigorous in your statistical analyses.
Results
The Authors listed a large number of diabetes-related sociodemographic characteristics, health-related characteristics as well as diabetes-related. I was expected to find how many people suffer from Type 1 and Type 2 Diabetes Mellitus. Please, add this aspect, it allows a clear comprehension of the results and the sample. Moreover, the Authors explained that this sample shows several comorbid conditions, I suggest the Authors add these conditions.
Conclusions
The Authors should add the clinical implication of their study from a psychological and medical point of view.
Through the whole manuscript, there are a lot of acronyms and this is fine but when you cite for the first time a word, you should write it entirely, such as BMI (in the Introduction section) and DFS-SF (in the Conclusion section).
Reviewer 2 Report
The manuscript by Mihyun Jeong “Predictors of Health-related Quality of Life in Korean Adults with Diabetes Mellitus” has been reviewed.
The manuscript aims to describe the association of health-related quality of life (HRQoL) with predictors of HRQoL in Korean adults with DM, using a sample from a nation wide study.
In general, the manuscript is easy to read. The research topic is of importance and scientifically sounded nevertheless, the manuscript is not transparent in terms of the origin of the data and the analytical decissions.
This reviewer has some suggestions.
- Abstract. Please provide numeric to describe the main association findings, Beta coefficients with 95CI.
Methods.
- Author stated in page 2 “ This sampling protocol employed a complex, stratified, multistage probability cluster sampling of the non-institutionalized civilian population of South Korea.” Please provide a brief description of the “complex, stratified, multistage probability cluster sampling” accompanied by previous publication on which it has been extensively explained.
- Please provide a full disclosure on how the data was obtained to conduct the present analysis, and, how the data can be requested for reproducibly purposes.
Results.
- Table 2 includes the numbers (n = 1,228, N = 2,175,029), where n = unweighted sample size; N = weighted sample size; what is the meaning of that? Do the author actually used data from the 2,175,029 ? Please clarify.
- Consider to add a flow chart diagraman to clarify the number and nature of participants actually included in the present study.
- Table 3 includes the note: “Note: n = unweighted sample size; N = weighted sample size” but such information is missing. Is that correct?
- Please provide a full rationality of using Complex-sample general linear regression analyses predicting HRQoL in people with DM.
- Author disclosed that: “This research did not receive any financial support.” Is that the case. Does the author has no relationship with the institution that generated a nation wide dataset? Is the dataset open for any researcher? Please clarify.
- Please replace the word gender by sex, as it is the assumption of this reviewer that authors of the data recorded the sex of the participants, but not their gender.
Reviewer 3 Report
Title and abstract:
The title synthesizes the main idea of the writing, it is explanatory by itself. In the same way it is concise and informative.
Introduction:
It incorporates the necessary information guiding the reader to identify the basic content of the paper quickly and to determine its relevance. It is semantically self-sufficient. Explains the purpose and aims of the text presented.
Point 1. Scientific background and rationale for the research being reported are correctly explained.
Point 2. The specific objectives are well established.
Methods:
Point 3. Study design and setting are correctly explained.
Point 4. Eligibility criteria and methods of selection of participants are correctly explained, but there is an inconsistency in the number of participants in the total sample: - Participants who completed the HRQoL questionnaire (n=1,438). The sample eligible for analysis consisted of 1,228 participants, after excluding participants with exclusion criteria (n=206). 1,438-206=1,232
Please, on line 70 delete diabetes mellitus and put only the abbreviation.
Point 5. What were the ranges established for monthly drinking and how the three categories of subjective health status were calculated?
Point 6. Variables, data sources/measurement and statistical methods are correctly.
Results:
Point 7. Participants, descriptive data, outcome data and main results are OK but, in Figure 1: Pain/discomfort is shown as significant (*) however it has p=0.12 and mobility has editing errors.
Point 8. Please, on line 217 delete health-related quality of life and put only HRQoL.
Point 9. Table 3: put in the foot correctly * p < 232 0.05, *** p < 0.01, *** p < 0.001. (on line 232).
Point 10. It would be useful to make a graph representing Table 3.
Discussion:
Point 11. Current and relevant article. As for discussion, they are correctly presented, resuming the conceptual guidelines.
Point 12. Please remove the "s" from word "its" on line 311, and change for "it is".
Point 13. You have a number of places where you spell out a word after you have abbreviated it or you abbreviate it after the first time you use it. That needs to be consistent.
Point 14. You use both the term sex and gender. Unless you are asking about gender identity, it should be sex.
Round 2
Reviewer 1 Report
Dear Authors,
the manuscript has been improved, therefore it is suitable for publication.